## Comment

psychology

pragmatics, infants, commentary

**Author for correspondence:**
Paula Rubio-Fernandez
e-mail: paula.rubio-fernandez@ifikk.uio.no

# Pragmatics for infants: commentary on Wenzel *et al.* (2020)

Paula Rubio-Fernandez[1], Victoria Southgate[2] and Ildikó Király[3]

[1]Department of Philosophy, University of Oslo, Norway
[2]Department of Psychology, University of Copenhagen
[3]Psychology Institute, Eötvös Loránd University, Budapest, Hungary

 PR-F, 0000-0003-1622-0967

## 1. Introduction

The *Sefo task* is an interactive paradigm designed to test false-belief (FB) reasoning in infants [1]. An experimenter (E1) shows two novel objects to a child, puts each object in a separate box and leaves the scene. Another experimenter (E2) then swaps the objects, either before E1 returns (FB condition) or while E1 is watching (true-belief (TB) condition). In the test phase, E1 points to one of the boxes and asks the child to retrieve the object for her. Southgate *et al.* [1] conducted three experiments with 17-month-old infants, which differed in how E1 phrased the request. In all three experiments, infants showed a reliable preference for the object in the non-referred box in the FB condition and the object in the referred box in the TB condition, suggesting sensitivity to E1's belief about the location of the object.

Recent studies have used the Sefo task, with mixed results [2–4]. Given the conflicting results of these studies, Wenzel *et al.* [5] conducted a collaborative study including a replication of Southgate *et al.* [1], plus two new versions of the Sefo task. Wenzel *et al.* only observed FB understanding in an adult control group, failing to replicate the original findings with 17-month-olds as well as 2- and 4-year-olds. They concluded that the Sefo task may not be a sensitive measure of FB understanding in children.

While all the above studies employed some version of the Sefo task, potentially important methodological differences may explain the discrepant results—a possibility that Wenzel *et al.* [5] also acknowledge. An exhaustive analysis of the protocols used in the Sefo studies is beyond the scope of this commentary. However, we address an important methodological question: what is *pragmatically important* in a FB task for infants?

## 2. Distinguishing 'direct' from 'conceptual' replications

One of the key manipulations in the Sefo task is the wording of the test question. The three experiments in the original study differed indeed by the phrasing that was used at test. For example, the prompt 'Do you remember what I put in here?', which was used in the first two experiments, was replaced by 'Do you know what's in here?' in the third experiment in order to avoid that children could circumvent FB reasoning by thinking only about the object that was originally there. Wenzel et al. [5] characterize their first experiment as a 'direct replication' of Southgate et al. [1]. However, the wording of the test phase did not correspond with the wording of any of the experiments in the original study, and it is, therefore, not accurate to describe it as a 'direct replication'.

Crucially, the test questions in Wenzel et al.'s Experiment 1 are not unproblematic: [While E1 is tapping on one of the boxes] 'Do you know what is in here? I want to play with this!' [Opening both boxes] 'Can you give it to me?' A literal interpretation of these probes suggest that E1 wants to play with the contents of the box she is tapping on, which she trusts the child is familiar with. Unlike in the original experiments, E1 in this version of the task does not make any reference to what she *expects* to find inside the box. In the original study, E1 either explicitly referred to the previous event (Remember what was inside this box?) or labelled the object (There's a sefo in this box!), thus providing evidence for what she thinks is in the box. The wording used by Wenzel and colleagues leaves open the possibility that E1 simply wants to play with whatever is inside that box. Thus, under a literal reading, the test questions in Wenzel et al.'s Experiment 1 directed the child towards the referred-to object, potentially compromising children's performance. This is an important deviation from the original design.

## 3. What is pragmatically felicitous for infants versus adults?

Wenzel et al. [5] designed two modified versions of the Sefo task that tried to make E1's motivation to request an object more reasonable, because the original design 'suffers from equivocality and pragmatic ambiguities' (p. 5). However, the rationale of the original Sefo task should also be considered from an infant's perspective: while it is true that E1 could have reached for the object herself, the Sefo task was designed as a *game between the experimenter and the infant*. As it normally happens when adults play with young children, the goal of the game need not be logical or transcendental: all that is often needed for the infant to engage is that the goal of the game be clear, and the exchange enjoyable.

Southgate et al. [1] ensured that the task procedure worked as a game by running a series of warm-up trials. E1 showed the infant two familiar objects, which the infant explored before E1 placed each object in a box. E1 then asked the child to find one of the objects, followed by the other, and continued this procedure until the child correctly chose the requested object twice in a row from two different boxes. Wenzel et al. [5] had problems ensuring that children passed the warm-up trials and relaxed both the inclusion criterion (bringing both objects needed not be in consecutive trials) and the set up (leaving the boxes open for some children). In the end, only nine children were tested on the original warm-ups, while 39 followed the modified procedure. This raises the question of whether these children understood their routine with E1 as a game. Engagement with the experimenter is known to modulate infants' willingness to help [6], and if children did not interpret the procedure as a game, it could also have led them to blindly trust the experimenter and consequently follow their ostensive pointing.

## 4. Concluding remarks

The Sefo task is a pragmatic task where infants need to understand the experimenter's referential intention in order to disambiguate their request. Since task performance depends on understanding referential intention (and not only false beliefs), the Sefo task is likely to be especially sensitive to two design features: the wording of the experimenter's request, and infants' motivation to employ their pragmatic abilities to override a default interpretation of the experimenter's pointing gesture. As things stand, there is conceptual replication [4] and non-replication [3] of the Sefo task, but some of the non-replications used alternative formulations of the experimenter's request [2,5]. In addition, Wenzel et al. [5] introduced further modifications to the warm-up trials, where infants were motivated

to engage in a game with the experimenter. Different versions of the Sefo task may, therefore, reveal how young children interpret the pragmatics of a communicative situation (e.g. game versus literal request) depending on verbal and contextual cues. Future Sefo studies should investigate this possibility by manipulating, in a controlled manner, not only the experimenter's beliefs about the location of the object, but also the pragmatics of the experimenter's request.

Data accessibility. This article has no additional data.
Competing interests. We declare we have no competing interests.
Funding. We received no funding for this study.

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
