## [Peer Review File · Royal Society Open Science]

Review History

Decision letter (RSOS-210247.R0)

Dear Dr Rubio-Fernandez

On behalf of the Editors, we are pleased to inform you that your Manuscript RSOS-210247 "Pragmatics for infants: Commentary on Wenzel *et al.* (2020)" has been accepted for publication in Royal Society Open Science subject to minor revision in accordance with the editor's request. Please the feedback from the Editor below my signature.

We invite you to respond to the comment and revise your manuscript. Below the Editors' comments we provide additional requirements. Final acceptance of your manuscript is dependent on these requirements being met. We provide guidance below to help you prepare your revision.

Please submit your revised manuscript and required files (see below) no later than 7 days from today's (ie 04-Mar-2021) date. Note: the ScholarOne system will 'lock' if submission of the revision is attempted 7 or more days after the deadline. If you do not think you will be able to meet this deadline please contact the editorial office immediately.

on behalf of Dr Teodora Gliga (Associate Editor) and Essi Viding (Subject Editor)
openscience@royalsociety.org

Associate Editor Comments to Author (Dr Teodora Gliga):

Comments to the Author:

This provides a very clear and constructive counterpoint to Wenzel et al. Please re-submit adding a short concluding paragraph (max 200 words).

===PREPARING YOUR MANUSCRIPT===

If you have been asked to revise the written English in your submission as a condition of publication, you must do so, and you are expected to provide evidence that you have received language editing support. The journal would prefer that you use a professional language editing service and provide a certificate of editing, but a signed letter from a colleague who is a native speaker of English is acceptable. Note the journal has arranged a number of discounts for authors

using professional language editing services
(<https://royalsociety.org/journals/authors/benefits/language-editing/>).

===PREPARING YOUR REVISION IN SCHOLARONE===

-- If you have uploaded ESM files, please ensure you follow the guidance at <https://royalsociety.org/journals/authors/author-guidelines/#supplementary-material> to include a suitable title and informative caption. An example of appropriate titling and captioning may be found at https://figshare.com/articles/Table_S2_from_ls_there_a_trade-

off_between_peak_performance_and_performance_breadth_across_temperatures_for_aerobic_sc
ope_in_teleost_fishes_/3843624.

Author's Response to Decision Letter for (RSOS-210247.R0)

See Appendix A.

Decision letter (RSOS-210247.R1)

Dear Dr Rubio-Fernandez,

It is a pleasure to accept your manuscript entitled "Pragmatics for infants: Commentary on Wenzel et al. (2020)" in its current form for publication in Royal Society Open Science. The comments of the reviewer(s) who reviewed your manuscript are included at the foot of this letter.

Please note that, as per our Comment and Reply policy, Comments are held at the Production stage to allow the authors of the original paper the opportunity to submit a Reply: <https://royalsociety.org/journals/ethics-policies/editorial-standards/>

on behalf of Dr Teodora Gliga (Associate Editor) and Essi Viding (Subject Editor)
openscience@royalsociety.org

Appendix A

DECISION LETTER

Dear Dr Rubio-Fernandez

On behalf of the Editors, we are pleased to inform you that your Manuscript RSOS-210247 "Pragmatics for infants: Commentary on Wenzel et al. (2020)" has been accepted for publication in Royal Society Open Science subject to minor revision in accordance with the editor's request. Please the feedback from the Editor below my signature.

[...]

Kind regards,

Anita Kristiansen
Editorial Coordinator

Comments to the Author:

This provides a very clear and constructive counterpoint to Wenzel et al. Please re-submit adding a short concluding paragraph (max 200 words).

RESPONSE LETTER

Dear Colleagues,

Thank you for the opportunity to revise and resubmit our commentary. We have followed your suggestion to add a 200-word concluding paragraph in order to end the commentary less abruptly.

Best wishes,
Paula Rubio-Fernandez